# Ultranarrow and Tunable Fano Resonance in Ag Nanoshells and a Simple Ag Nanomatryushka

**DOI:** 10.3390/nano11082039

**Published:** 2021-08-10

**Authors:** Ping Gu, Xiaofeng Cai, Guohua Wu, Chenpeng Xue, Jing Chen, Zuxing Zhang, Zhendong Yan, Fanxin Liu, Chaojun Tang, Wei Du, Zhong Huang, Zhuo Chen

**Affiliations:** 1Institute of Advanced Photonics Technology, College of Electronic and Optical Engineering & College of Microelectronics, Nanjing University of Posts and Telecommunications, Nanjing 210023, China; guping@njupt.edu.cn (P.G.); 1219023112@njupt.edu.cn (X.C.); 1219023306@njupt.edu.cn (G.W.); cp_xue@njupt.edu.cn (C.X.); jchen@njupt.edu.cn (J.C.); zxzhang@njupt.edu.cn (Z.Z.); 2College of Science, Nanjing Forestry University, Nanjing 210037, China; zdyan@njfu.edu.cn; 3College of Science, Zhejiang University of Technology, Hangzhou 310023, China; lfx63@163.com; 4College of Physics Science and Technology, Yangzhou University, Yangzhou 225002, China; wdu@yzu.edu.cn; 5College of Physics and Electronic Engineering, Jiangsu Second Normal University, Nanjing 210013, China; huangzhong89@126.com; 6National Laboratory of Solid State Microstructures, College of Physics, Nanjing University, Nanjing 210093, China

**Keywords:** Ag nanoshell, Ag nanomatryushka, Fano resonance, tunable

## Abstract

We study theoretically the Fano resonances (FRs) produced by the near-field coupling between the lowest-order (dipolar) sphere plasmon resonance and the dipolar cavity plasmon mode supported by an Ag nanoshell or the hybrid mode in a simple three-layered Ag nanomatryushka constructed by incorporating a solid Ag nanosphere into the center of Ag nanoshell. We find that the linewidth of dipolar cavity plasmon resonance or hybrid mode induced FR is as narrow as 6.8 nm (corresponding to a high *Q*-factor of ~160 and a long dephasing time of ~200 fs) due to the highly localized feature of the electric-fields. In addition, we attribute the formation mechanisms of typical asymmetrical Fano line profiles in the extinction spectra to the constructive (Fano peak) and the destructive interferences (Fano dip) arising from the symmetric and asymmetric charge distributions between the dipolar sphere and cavity plasmon or hybrid modes. Interestingly, by simply adjusting the structural parameters, the dielectric refractive index required for the strongest FR in the Ag nanomatryushka can be reduced to be as small as 1.4, which largely reduces the restriction on materials, and the positions of FR can also be easily tuned across a broad spectral range. The ultranarrow linewidth, highly tunability together with the huge enhancement of electric fields at the FR may find important applications in sensing, slow light, and plasmon rulers.

## 1. Introduction

Engineering Fano resonance (FR) in plasmonic nanostrucures and artificial metamaterials has become an important research focus in recent years due to its wide applications such as surface-enhanced Raman scattering (SERS) [1], refractive-index sensing or biosensing sensors [2,3,4,5,6,7], plasmon rulers [8], molecular identification [9], slow-light devices [10], narrow-band absorbers [11], nonlinear optical processes [12,13], and so on. FR primally charcterized by the intrinsic sharp asymmetric line-shape in the far-field spectrum and the large electromagnetic field enhancement in the near-field, which results from the constuctive and destructive far-field interferences with strong interacitons between the narrow subradiant (dark) modes and the broad superradiant (bright) plasmon resonances [14,15,16,17].

Up to now, various plasmonic structures have been proposed and demonstrated to generate FR in a wide spectral range from THz to the optical region [14,15,16,17,18,19,20], such as dolmen structures [21,22,23], nanoparticles assemblies [24,25,26,27,28], concentric [29,30,31,32,33,34] and nonconcentric ring/disk cavities [35,36], and metallic nanoparticle/graphere hybrid sturctures [37,38]. In addition, the introduction of asymmetry into metallic nanostructures enables the generation of new dark (narrow) plasmon modes, and has been demonstated to be an effective strategy to engineering the line-shape, the linewidth, and the numbers of FR (single, double, triple and so on) [35,39,40,41]. Espectially, the excitation of multiple FRs can largely promote the accuracy and performance of plasmon ruler and nano-sensor [8,28]. In recent years, the tunability of the FR whose resonant wavelength and excitation strenghth can be adjusted by internal or external parameters has also attracted extensive research attentions, because the tunable FR can largely extend the functionalities of plasmonic metamaterials in practice applications. Tuning FR by directly adjusting the structural and material parameters of plasmonic nanostructures has been proved to be an effective approach, which is a passive way [42,43,44]. More recently, FRs in plasmonic nanostrcutures have also been demonstrated to be actively tuned by changing the external parameters such as voltages [45,46], temperature [47], mechanical stress [48], magnetic field [49], and optical parameters [50,51].

Unlike the metallic nanoparticle assemblies and asymmetry nanostructures, to generate FR with high complexity and high precisely (inter-particle separation on the order of several tens of nanometers) [24,25,26,27,28,35,36,39,40,41], a nobel metal (Au/Ag) nanoshell composed of a spherical dielectric core and a metallic shell is the simplest plasmonic nanostructure [52]. More recently, these have been demonstrated both theoretically and experimentally to be capable of supporting multiple sharp FRs induced by the multipolar high-*Q* cavity plasmon modes when the dielectric core size can be comparable to the wavelength of light (beyond the electrostatic limit) [53,54,55,56]. However, the intensity of the FR caused by the lowest-order cavity plasmon mode in a metallic shell is relatively small, both in the theoretical prediction and the experimental observation, which is probably due to the mismatch of resonant energies between sphere and cavity plasmon modes caused by the low refractive index of the dielectric core [56].

In this paper, we investigate theoretically the formation mechanisms and the tunability of the FRs in the Ag nanoshell and the Ag nanomatryushka formed by incorporating a solid silver nanosphere into the center of the same Ag nanoshell. We find that the resonant energies of the lowest-order (dipolar) cavity plasmon mode in the Ag nanoshell or the hybrid resonance in the Ag nanomatryushka are highly concentrated within the dielectric core or within the dielectric layer, which leads to the formation of FRs with the linewidths as narrow as 6.8 nm, corresponding to a high *Q*-factor of ~160 and a long dephasing time of ~200 fs. We further demonstrate that the typical asymmetrical Fano line profiles in the extinction spectra are due to the constructive (Fano peak) and destructive interferences (Fano dip) with the symmetric and asymmetric charge distributions between the dipolar sphere and cavity plasmon or hybrid modes. Interestingly, the dielectric refractive index required for the strongest FR in the Ag nanomatryushka can be reduced to be as small as 1.4 by simply adjusting the structural parameters, which is much smaller than that of the Ag nanoshell (*n*~2.6). In addition, the resonant positions of FR in the Ag nanomatryushka can also be easily tuned across a broad spectral range.

## 2. Methods

Figure 1 schematically displays the Ag nanoshell (see Figure 1a) and a simple three-layered Ag nanomatryushka (see Figure 1b) to be investigated in this paper. The Ag nanoshell consists of a dielectric nanosphere core (radius: *r*_core1_, refractive index: *n*_core1_) wrapped by a silver shell with thickness of *t* (see Figure 1a). The simple three-layered Ag nanomatryushka is constructed by incorporating a solid Ag nanosphere (radius: *r*_core2_) into the center of the same Ag nanoshell. The dielectric thickness and dielectric refractive index in the Ag nanomatryushka are *d* and *n*_d_, respectively (see Figure 1b). As a result, the total sizes of the Ag nanoshell and the Ag nanomatryushka are equal (*r*_core1_
*+ t* = *r*_core2_ + *d + t*). Noting that the silver shell thickness (*t*) in this study is intentionally chosen and fixed to be ~50 nm, which is far beyond the optical skin depth of ~20 nm for silver at the visible and near-infrared spectral range, so that the Ag nanoshell and Ag nanomatryushka can effectively act as the closed metallic cavity, and finally leads to a good light confinement ability and thus a considerable narrow resonant mode [53,56].

Due to the perfect spherical symmetry of the proposed Ag nanoshell and the Ag nanomatryushka (multilayered sphere), the extinction, scattering and absorption properties produced by the interaction between the plane wave and the proposed nanostructures can be solved analytically using an improved recursive algorithm based on Mie scattering theory [57,58]. In the analytical Mie solutions, the total extinction efficiency *Q*_ext_ defined as extinction cross section divided by the cross section of the proposed nanoparticle is expressed as the following equation:(1)Qext=2k2r2 ∑l(2l+1)[Re(al)+Re(bl)]
where *k* is the wavenumber, *r* is the outer radius of the proposed nanoparticle, *a_l_* and *b_l_* is the *l*-th (*l* represents the angular mode number) order transverse-magnetic and transverse-electric Mie scattering coefficient, respectively. The electromagnetic fields in different regions can be expanded in terms of complex spherical eigenvectors, and the expansion coefficients can be obtained by the boundary conditions [58]. The permittivity of the silver shell is expressed by a Drude mode: *ε*_Ag_ = *ε*_∞_ − $\omega_{p}^{2}$/[*ω*(*ω* + *iω*_c_)], where the parameters (*ε*_∞_ = 3.8355, *ω*_p_ = 1.3937 × 10^16^ rad/s, and *ω*_c_ = 2.9588 × 10^13^ rad/s) of the Drude model are obtained by fitting the experimental date of Johnson and Christy [59]. The outer medium is assumed to be air with refractive index of 1.0.

Up to now, small-sized spherical metal/dielectric/metal nanomatryushkas have been successfully fabricated using chemical methods [29,30,32,33,34]. However, to fabricate large-sized spherical nanomatryushkas using the chemical method it is still a challenging task. More recently, with the development of micro/nanofabricating technology, near-perfect metallic shells with a large dielectric core size have been successfully fabricated by using the self-supporting method [56], which can provide a possible method for realizing the proposed spherical nanomatryushka. In brief, the silver nanosphere array is firstly self-assembled at the water/air interface by using the modified Langmuir-Blodgett method [60], and then transferred onto the substrates with through-holes to form a self-supporting layer. Secondly, the dielectric and metal materials with the precisely controlled thickness are physically deposited onto the upper and lower surface of the as-prepared self-supporting silver nanosphere array, and finally constructed the three-layered core-shell nanoparticles (nanomatryushka). 

## 3. Results and Discussion

Figure 2a displays the extinction efficiency spectra calculated from the Mie scattering coefficient of a_1_ for an Ag nanoshell (red line, *r*_core1_ = 150 nm, *n*_core1_ = 1.0) and a simple Ag nanomatryushka (blue line, *r*_core2_ = 80 nm, *d* = 70 nm, *n*_d_ = 1.0), respectively. For comparison, Figure 2a also presents the extinction efficiency spectra for a solid Ag nanosphere with the identical size as total Ag nanoshell and Ag nanomatryushka (black dash line, *r* = 200 nm). It is clearly seen from Figure 2a that the proposed three plasmonic nanostructures all show a broad extinction peak with the same resonant wavelength (*λ*_A_ = 1095 nm), which is marked as resonance A. In addition, an additional narrow resonance for both the Ag nanoshell and the Ag nanomatryushka is appeared at the shorter wavelength, which is marked as resonance B (*λ*_B_ = 436 nm) and C (*λ*_C_ = 646.6 nm), respectively.

In order to further understand the resonances A, B, C supported by the proposed plasmonic nanostructures, we determined the near-field profiles at the selected wavelength using analytical Mie solution [57,58]. Figure 2b–d show the electric-field intensity (|*E*/*E*_0_|^2^) distributions calculated at the resonances A (*λ*_A_ = 1095 nm), B (*λ*_B_ = 436 nm), C (*λ*_C_ = 646.6 nm), respectively. It is clearly seen from Figure 2b that the electric fields for resonance A supported by the solid Ag nanosphere are mainly concentrated near the outer surface of the silver shell. This distribution characteristic reveals the broad spectral feature of resonance A (see Figure 2a), because the resonant energy is easily radiated into the air (large radiative loss). In addition, we have also calculated the corresponding electric fields of the Ag nanoshell and the Ag nanomatryushka at the same resonant wavelength, which presents the same field pattern as the solid Ag nanosphere [not shown here]. Therefore, such a broad Mie mode corresponds to the excitation of dipolar sphere plasmon mode [53,54]. In contrary, the electric fields of resonance B are highly concentrated within the dielectric core of the Ag nanoshell as shown in Figure 2c, which corresponds to the excitation of dipolar cavity plasmon mode [53,55]. This highly localized feature of electric fields reveals the narrow linewidth feature of cavity plasmon mode due to the low radiative loss.

For resonance C supported by the Ag nanomatryushka, the electric fields have the combined features of both the sphere plasmon mode and the cavity plasmon resonance (see Figure 2d). Specifically, the electric fields of resonance C have similar distribution pattern as the sphere plasmon mode, and are also highly concentrated within the dielectric layer (the feature of cavity plasmon resonance). As a result, the resonance C not only has a strong near-field enhancement (1535-fold, see Figure 2d), but also maintains the characteristic of narrow linewidth in the far-field spectrum (see Figure 2a). It should be pointed out that the resonant properties of mode C (hybrid) in the Ag nanomatryushka apparently arise from the hybridization between the sphere plasmon mode supported by the solid Ag nanosphere and the cavity plasmon resonance supported by the Ag nanoshell.

Next, we focus on the sharp FR arising from the near-field interaction between the narrow cavity plasmon or hybrid mode and the broad sphere plasmon resonance. For this purpose, we should tune the narrow-band cavity plasmon or hybrid mode across the broad extinction peak (sphere plasmon resonance). Because the electric fields of the cavity plasmon or hybrid resonance are highly confined within the dielectric core in the Ag nanoshell or dielectric layer in the Ag nanomatryushka (see Figure 2), the resonant wavelengths of these two modes can thus be easily tuned by varying the refractive index of dielectric core (*n*_core1_) or dielectric layer (*n*_d_). Figure 3a displays the extinction efficiency spectra (*a*_1_) for the Ag nanoshell (*r*_core1_ = 150 nm) with the dielectric core index (*n*_core1_) increasing from 1.0, 1.6, 2.2, 2.6, 3.0, 3.6 to 4.0. With increasing the *n*_core1_, it is evident that the resonant wavelengths of the sphere plasmon modes (broad extinction peaks) are almost unchanged, since the electric fields of this mode are mainly distributed in the outer surface of the silver shell, and are thus not perturbed by the change in the core medium. However, as *n*_core1_ is increased from 1 to 2.2, the cavity plasmon mode is quickly red-shifted and approaches to the extinction peak (sphere plasmon mode) from the shorter-wavelength side. The most obvious spectral change in this process is that the intensity of extinction dip is increased quickly (see Figure 3a). Similarly, the intensity of extinction peak is increased quickly as the cavity plasmon mode approaches to the extinction peak (sphere plasmon mode) from the longer-wavelength side, in which the *n*_core1_ decreases from 4.0 to 3.0 (see Figure 3a). As a result, there appears a strongest asymmetrical FR with both the strong extinction dip and the strong extinction peak when the cavity plasmon mode approaches the extinction peak at *n*_core1_ = 2.6 (red line in Figure 3a). The evolution process of the extinction efficiency spectrum in the Ag nanomatryushka is similar to the Ag nanoshell as the *n*_d_ increases from 1.0 to 4.0 as shown in Figure 3b. Meanwhile, the strongest FR occurs at a value of *n*_d_ = 1.8, which is much smaller than that of the Ag nanoshell (*n*_core1_ = 2.6) (red line in Figure 3b). It should be pointed out that this value (*n*_d_) can be further reduced by simply adjusting the structural parameters in the Ag nanomatryushka, which will be discussed in the following.

FR is a well-known interference phenomenon in atomic physics, where the interference of a discrete autoionized state with a continuum gives rise to characteristically asymmetric peaks in extinction spectra as shown in Figure 4a [61]. Figure 4b schematically displays the hybridization between the sphere plasmon mode supported by a solid Ag nanosphere and the cavity plasmon mode supported by a metallic (Ag) cavity, resulting in the formation of a symmetric and antisymmetric hybrid plasmon modes supported by the Ag nanoshell. In the Ag nanoshell and the Ag nanomatryushka, the bandwidths of the highly localized cavity plasmon and hybrid modes are narrow and are regarded as the subradiant “dark” modes due to the weak radiation damping. On the contrary, the bandwidth of the sphere plasmon resonance is very broad and is regarded as a superradiant “bright” mode due to the quick radiation damping. The subradiant dark mode and the superradiant bright mode can be analogic to a discrete autoionized state with a continuum in atomic physics, respectively [14,15,16,17,61]. As a result, FRs arise from the interactions between the two kinds of plasmon modes in the Ag nanoshell and the Ag nanomatryushka, and they are characterized by an asymmetrical Fano line-profile in the extinction spectra.

For a better understanding of the observed FRs, the calculated extinction efficiency spectrum is fitted in the vicinity of the strongest FR using a Fano formula: *F*(*ε*) = *σ*_bg_ + *σ*_0_(*ε* + *q*)^2^/(1 + *ε*^2^), where *σ*_bg_ and *σ*_0_ are the background and normalized extinction, *ε* = 2(*λ* − *λ*_res_)/*Г* with *λ*_res_ and *Г* being the resonant position and linewidth of the cavity plasmon or hybrid modes, respectively. Figure 5a,d display two examples of such a Fano fitting for the cavity plasmon mode induced FR in the nanoshell and the hybrid mode-induced FR in the Ag nanomatryushka, respectively. The fitted curves (olive lines) are in good agreement with the theoretical spectra (red line). From the precisely fitting, the resonant wavelengths (*λ*_res_), linewidths (*Г**/τ*) of the cavity plasmon and hybrid resonances are 1085 nm, 6.8455 nm (7.21 meV) and 1145.6 nm, 6.7881 nm (6.41 meV), respectively (see Figure 5a,d). With the extracted resonant wavelengths (*λ*_res_), and linewidths (*Г**/τ*), the *Q*-factor and the dephasing time (*T*) are further calculated by using the following equations [62]:*Q = λ_res_/Г*(2)
*T = 2ℏ/τ*(3)
where *ħ* = 6.582119514 × 10^−16^ eV·ns. From Equations (2) and (3), the *Q*-factors of cavity plasmon and hybrid modes are as high as 158.5 and 168.8, which corresponds to a long dephasing time of 182.6 fs and 205.4 fs, respectively. Although both the cavity plasmon and hybrid modes have large *Q*-factors and long dephasing times, their effective mode volumes are quite different. The normalized effective mode volume is calculated according to the following equation by using the three-dimensional finite element method software COMSOL Multiphysics [63]:(4)$$V_{\mathrm{eff}}^{N}=[\frac{\int_{V}^{}{\varepsilon (r)|E(r)|}^{2}\mathrm{dV}}{{\max[\varepsilon (r)|E(r)|}^{2}]}]∕(\frac{\lambda }{n}{)}^{3}$$
where *ε*(*r*) is the dielectric constant, |*E*(*r*)| is the electric field strength, and *V* is the vol ume encompassing the cavity with a boundary in the radiation zone of the cavity plasmon or hybrid mode [63]. According to the above expression, the numerically calculated normalized effective mode volume of FR in Figure 5a,d is 3.6 × 10^−^^2^ and 2.06 × 10^−^^3^, respectively. Therefore, the normalized effective mode volume of hybrid mode is about 17.5 times smaller than that of the cavity plasmon resonance, revealing that the hybrid mode in Ag nanomatryushka could be more beneficial for light-mater interaction [63].

To get more insight into the underlining physical mechanisms of the FRs, we plot in Figure 5b,c and Figure 5e,f the electric-field vectors on the *k-E* plane for the Fano dip (peak) *I*, and Fano dip (peak) *II* indicated by the arrows in Figure 5a and Figure 5d, respectively. According to the distribution of the electric-field vectors, we have also deduced the corresponding distributions of the positive and negative charges induced on the inner/outer surface of the silver shell and the surface of the solid Ag nanosphere. For the Fano dip *I* at 1082.4 nm (see Figure 5b), it is evident that the strong near-field coupling between the dipolar sphere and cavity plasmon modes leads to the formation of a hybridized plasmon resonance with asymmetric charge distributions on the outer/inner surface of the silver shell. This means that two dipolar plasmon modes (sphere and cavity plasmon modes) oscillate out of phase, and therefore their dipolar radiations will interfere destructively in the far-field region, which is signaled by a Fano dip in the extinction spectra (see Figure 5a). In contrast, for the Fano peak *I* at 1089.5 nm, the interaction between the dipolar sphere and cavity plasmon modes results into a hybrid plasmon mode with symmetric charge distributions on the out/inner surface of the silver shell, as displayed in Figure 5c. In this case, the two dipolar plasmon modes outside and inside the Ag nanoshell oscillate in phase, so their dipolar radiations will interfere constructively in the far-field region, and thus appears a Fano peak in the extinction spectra (see Figure 5a).

Differs from the Ag nanoshell, the electrical dipolar outside (sphere plasmon mode) the Ag shell can induce two in-phase dipoles inside the cavity in the Ag nanomatryushka as shown in Figure 5e,f. However, for the Fano dip *II* at 1142.6 nm, the asymmetric charge distributions between the sphere plasmon resonance and the hybrid mode (see Figure 5e) can still lead to the interfere destructively in the far-field region, and thus a Fano dip appears in the extinction spectrum (see Figure 5d). For the Fano peak *II* at 1149.6 nm, the symmetric charge distributions between the sphere plasmon resonance and the hybrid mode are displayed in Figure 5f, resulting in the interfere constructively in the far-field region, and thus showing a Fano peak in the extinction spectrum (see Figure 5d). Here, we should mention that such a theoretical model of interacting dipolar oscillations have also been applied to explain successfully the FR observed in other classical systems [14,15,16,17].

In the following, we focus on the tunability of the FRs. As has been demonstrated above, adjusting the dielectric refractive index is an effective strategy for tuning the FR (see Figure 3). However, the relatively high refractive index where the strongest FR happening in Ag nanoshell (*n*_core1_ = 2.6) is limited by only several semiconductor materials [64]. At this point, the Ag nanomatryushka only needs lower refractive index (*n*_d_ = 1.8) compared with the Ag nanoshell (see Figure 3). 

It should also be pointed out that this value (*n*_d_) required for the strongest FR can be further reduced from 1.8 to 1.4 by simply adjusting the *r*_core2_ (the radius of solid Ag nanosphere) from 80 nm (*d* = 70 nm), 87 nm (*d* = 63 nm), 94 nm (*d* = 56 nm), 101 nm (*d* = 49 nm) to 108 nm (*d* = 42 nm) in the Ag nanomatryushka as shown in Figure 6a, which largely reduces the restriction on materials. In this work, we choose the refractive indexes of dielectric layer (*n*_d_ = 1.8, 1.7, 1.6, 1.5, and 1.4) only for the theoretical demonstration of the relationship between the *n*_d_ required for the strongest FR and the structural parameter of the proposed Ag nanomatryushka, which can provide valuable guidance for experiments. In other words, we can choose appropriate structural parameters (the size of the Ag nanosphere and the thickness of the dielectric layer) according to the actual dielectric materials used in the experiments to realizing the strongest FR. For example, the refractive index of Al_2_O_3_ material in the near-infrared spectral range is 1.75, which is close to the theoretical value of 1.8. In addition, the FRs can also be tuned across the broad extinction peak from 900 nm to 1365 nm by increasing the *r*_core2_ from 50 nm (*d* = 100 nm), 60 nm (*d* = 90 nm), 70 nm (*d* = 80 nm), 80 nm (*d* = 70 nm), 90 nm (*d* = 60 nm) to 100 nm (*d* = 50 nm) as displayed in Figure 6b, in which the *n*_d_ is fixed to be 1.8. The above results apparently demonstrate the highly tunable of the FR in Ag nanomatryushka.

## 4. Conclusions

In summary, we have investigated theoretically the FRs resulting from the interaction between wide-band dipolar plasmon sphere resonance and narrow-band dipolar cavity plasmon mode supported by the Ag nanoshell or the hybrid mode in the Ag nanomatryushka. The highly localized feature of the electric-fields for both the dipolar cavity plasmon mode and the hybrid resonance leads to the formation of FR with linewidth as narrow as 6.8 nm, which corresponds to a high *Q*-factor of ~160 and a long dephasing time of ~200 fs. The constructive and destructive interferences in the far-field dipolar radiations associated with the symmetric and asymmetric charge distributions in the near-field between the dipolar sphere and cavity plasmon or hybrid resonance result in the appearance of asymmetrical Fano line-profiles in the extinction spectra. Compared with the Ag nanoshell, the dielectric refractive index required for the strongest FR can be reduced to be as small as 1.4, and the resonant positions can be tuned across a broad spectral range only by simply adjusting the structural parameters in the Ag nanomatryushka. The ultranarrow linewidth, high *Q*-factor, long dephasing time, small mode volume and highly tunability accompanied by a huge enhancement of electric fields at the FR of our proposed Ag nanomatryushka may be used for some applications such as sensing, slow light, and plasmon ruler.

## Figures and Tables

**Figure 1 nanomaterials-11-02039-f001:**
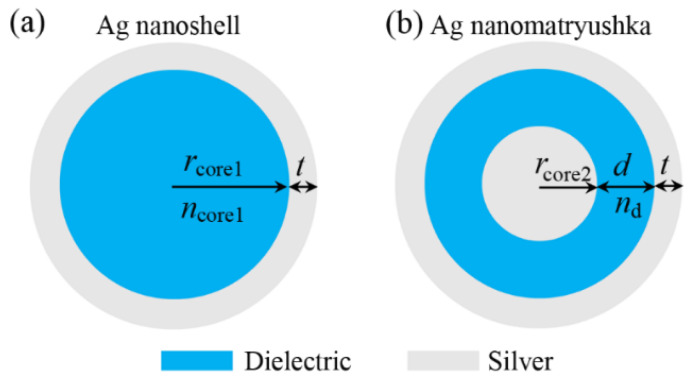
(**a**) Schematic of an Ag nanoshell (**a**) and a simple three-layered Ag nanomatryushka (**b**). The Ag nanoshell is formed by a silver layer (thickness: *t*) coating a dielectric nanosphere (radius: *r*_core1_; refractive index: *n*_core1_), and the Ag nanomatryushka is constructed by incorporating a solid Ag nanosphere (radius: *r*_core2_) into the center of the same Ag nanoshell as in (**a**). The dielectric thickness and dielectric refractive index in the Ag nanomatryushka are *d* and *n*_d_, respectively.

**Figure 2 nanomaterials-11-02039-f002:**
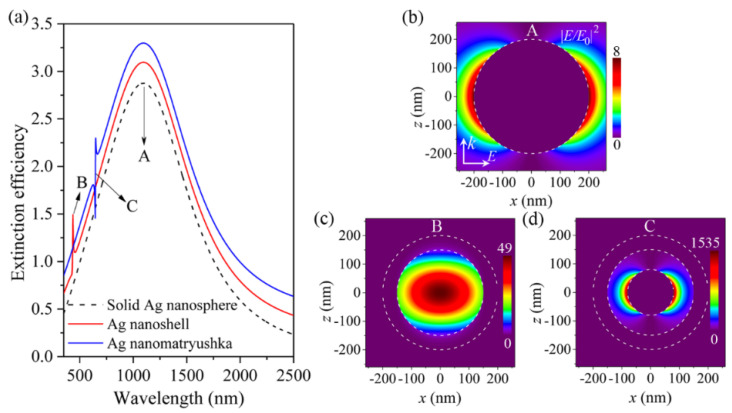
(**a**) Calculated extinction efficiency spectra (*a*_1_) for a solid Ag nanosphere (*r* = 200 nm, black dash line), an Ag nanoshell (*r*_core1_ = 150 nm, *n*_core1_ = 1.0, red line) and an Ag nanomatryushka (*r*_core2_ = 80 nm, *d* = 70 nm, *n*_d_ = 1.0, blue line), respectively (each spectrum vertically offset by 0.2 for clarity). (**b**–**d**) Electric field intensity distributions at *k-E* plane for resonances A, B, C in (**a**), respectively.

**Figure 3 nanomaterials-11-02039-f003:**
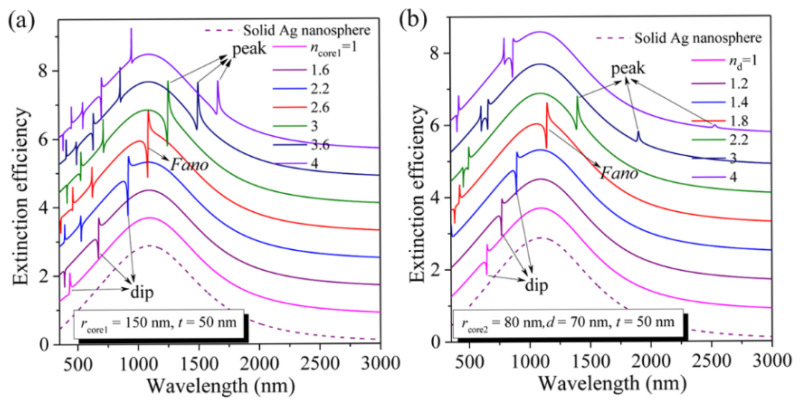
(**a**) Calculated extinction efficiency spectra (*a*_1_) for an Ag nanoshell (*r*_core1_ = 150 nm) with different *n*_core1_ from 1.0 to 4.0 (each spectrum vertically offset by 0.8 for clarity). (**b**) Calculated extinction efficiency spectra (*a*_1_) for an Ag nanomatryushka (*r*_core2_ = 80 nm, *d* = 70 nm) with different *n*_d_ from 1.0 to 4.0 (each-spectrum vertically offset by 0.8 for clarity).

**Figure 4 nanomaterials-11-02039-f004:**
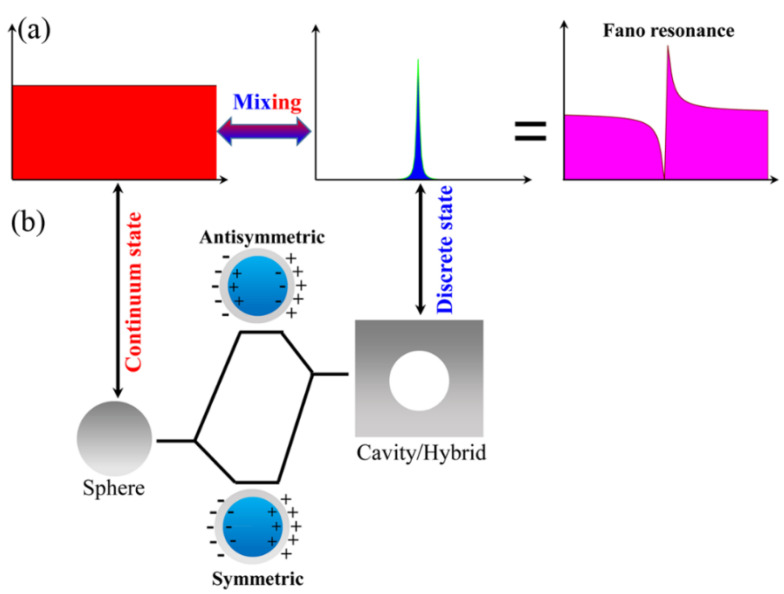
(**a**) Formation of the FR as a superposition of the Lorentzian line shape of discrete state with a flat continuous background. (**b**) Schematic of the plasmon hybridization. The hybridization of a sphere and a cavity plasmon leads to the formation of a symmetric and antisymmetric plasmon resonances of the Ag nanoshell.

**Figure 5 nanomaterials-11-02039-f005:**
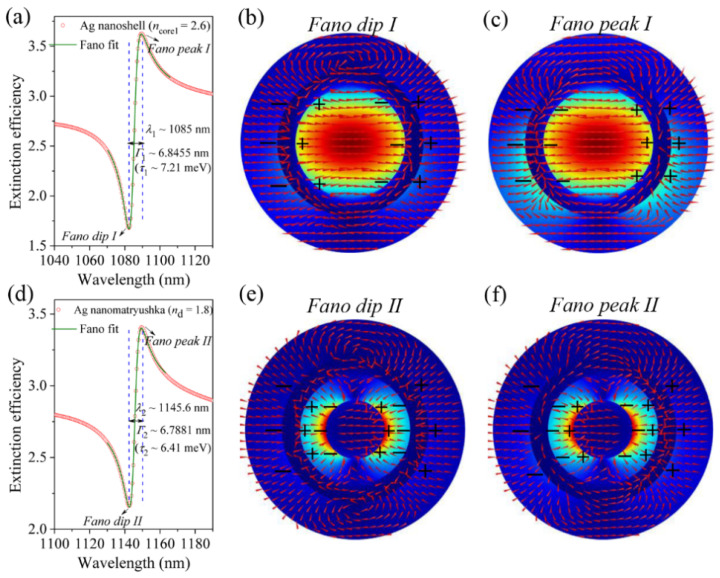
(**a**,**d**) Fano fitting (olive lines) the calculated extinction efficiency spectra (*a*_1_) for an Ag nanoshell (*r*_core1_ = 150 nm, *n*_core1_ = 2.6) and an Ag nanomatryushka (*r*_core2_ = 80 nm, *d* = 70 nm, *n*_d_ = 1.8), respectively. (**b**,**c**,**e**,**f**) show the electric-field vector distributions of the dipolar cavity plasmon and the hybrid modes at the Fano dip (peak) *I* and Fano dip (peak) *II* labeled in (**a**) and (**d**), respectively. Red arrows represent field direction and colors show field intensity. Black signs “+” and “−” stand for positive and negative charges, respectively.

**Figure 6 nanomaterials-11-02039-f006:**
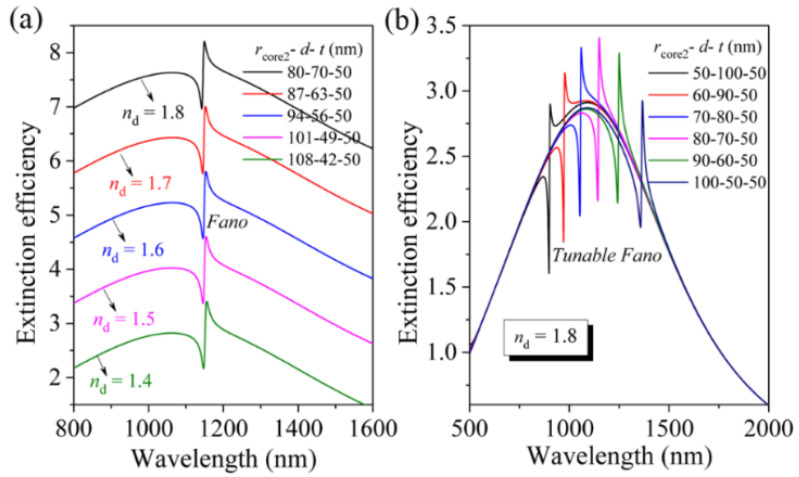
Tunable FR in the Ag nanomatryushka. (**a**) Calculated extinction efficiency spectra (*a*_1_) for an Ag nanomatryushka with different *r*_core2_, *n*_d_ and *d* (other parameters: *r*_core2_ + *d* = 150 nm) (each spectrum vertically offset by 1.2 for clarity). (**b**) Calculated extinction efficiency spectra (*a*_1_) for an Ag nanomatryushka with different *r*_core2_ and *d* (other parameters: *n*_d_ = 1.8, and *r*_core2_ + *d* = 150 nm).

## Data Availability

The study did not report any data.

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
