# Peer review of "Ultranarrow and Tunable Fano Resonance in Ag Nanoshells and a Simple Ag Nanomatryushka"

_nanomaterials, 2021, doi:10.3390/nano11082039_

Round 1

Reviewer 1 Report

In this work, the spectral properties and excitation of Fano lineshapes through the use of plasmonic nanomatryushka structures have been discussed and presented. Using numerical studies, it is shown that the tailored ring-disk configuration sustains significantly narrow Fano lineshapes. Although the work comprehensively analyzed the properties of the system and demonstrated the excitation of pronounced dark lineshapes, the manuscript suffers from important drawbacks. I listed my comments below and will suggest the authors to address the comments carefully in the revised version of the manuscript.

1) Since the work is prepared based on simulation investigations, there must be a separate section to explain the important settings of the conducted method by having stress on the employed tool. What type of analysis was conducted to extract the results? FEM or FDTD or something else?

2) There are missed works in the literature that focused on the spectral properties of concentric nanodisk-ring configurations, which must be considered in the bibliography, such as: Nano Letters 15(10), 6419-6428 (2015), Plasmonics 11(2), 493-501 (2016), The Journal of Physical Chemistry C, 125(3), 1865-1873, and many other works.

3) The authors must plot the energy level diagram of the system to demonstrate the hybridization of excited states and interference of dark and bright modes.

4) The spectral linewidth or Q-factor, dephasing time, and mode volume of the induced Fano features must be quantified accurately. These parameters play fundamental role in the evaluation of the performance. Several methods have been proposed for the calculation of those parameters and the following articles may help the authors:  Nanoscale 11(17), 8091-8095 (2019), Advanced Optical Materials 8(16), 1902025 (2020).

Reviewer 2 Report

This manuscript deals with Fano resonance (FR) effects in a Ag nanoshell and Ag nanomatryushka. Since the conventional Ag nanoshell has low FR intensity, the authors proposed a simple three layer Ag nanomatryushka. The simulation results reveal that the proposed structure has very narrow FR peaks due to the highly localized E-field. However, the reviewer is skeptical on the design (nanomatryushka) that the authors proposed, because it is extremely hard to generate those structures. Moreover, in figure 5, the authors used materials with refractive index of 1.8, which is impractical in real world. Without some of the preliminary experimental results, it is not acceptable in the present form. 

Round 2

Reviewer 1 Report

The work is acceptable as is.

Reviewer 2 Report

The revised version is well written based on the reviewer's concerns.